# Measurement report: Spatiotemporal and policy-related variations of PM$_{2.5}$ composition and sources during 2015-2019 at multiple sites in a Chinese megacity

Xinyao Feng[1], Yingze Tian[1,2*], Qianqian Xue[1], Danlin Song[3], Fengxia Huang[3], and Yinchang Feng[1]

[1]State Environmental Protection Key Laboratory of Urban Ambient Air Particulate Matter Pollution Prevention and Control, College of Environmental Science and Engineering, Nankai University, Tianjin, 300071, China
[2]CMA-NKU Cooperative Laboratory for Atmospheric Environment-Health Research (CLAER/CMA-NKU)
[3]Chengdu Research Academy of Environmental Sciences, China

*Correspondence to*: Yingze Tian (tianyingze@hotmail.com; 015058@nankai.edu.cn)

**Abstract.** A thorough understanding of the relationship between urbanization and PM$_{2.5}$ (fine particulate matter with aerodynamic diameter less than 2.5 μm) variation is crucial for researchers and policymakers to study health effects and improve air quality. In this study, we selected a rapidly developing Chinese megacity, Chengdu, as the study area to investigate the spatiotemporal and policy-related variations of PM$_{2.5}$ composition and sources based on long-term observation at multiple sites. A total of 836 samples were collected from 19 sites in winter, 2015-2019. According to the specific characteristics, 19 sampling sites were assigned to three layers. Layer 1 was the most urbanized area referred to the core zone of Chengdu, layer 2 was located in the outer circle of layer 1, and layer 3 belonged to the outermost zone with the lowest urbanization level. The average PM$_{2.5}$ concentrations for five years were in the order of layer 2 (133 μg m$^{-3}$) > layer 1 (126 μg m$^{-3}$) > layer 3 (121 μg m$^{-3}$). Spatial clustering of the chemical composition at the sampling sites was conducted for each year. The PM$_{2.5}$ composition of layer 3 in 2019 was found to be similar to that of the other layers two or three years ago, implying that urbanization levels had a strong effect on air quality. During the sampling period, a decreasing trend was observed for the annual average concentration of PM$_{2.5}$, especially at sampling sites in layer 1, where the stricter control policies were implemented in layer 1. The SO$_4^{2-}$/NO$_3^-$ mass ratio at most sites exceeded 1 in 2015 but dropped to less than 1 since 2016, reflecting decreasing coal combustion and increasing traffic impacts in Chengdu, and can be further supported by temporal variations of the SO$_4^{2-}$ and NO$_3^-$ concentrations. The positive matrix factorization (PMF) model was applied to quantify PM$_{2.5}$ sources. A total of five sources were identified, with average contributions of 15.5% (traffic emissions), 19.7% (coal and biomass combustion), 8.8% (industrial emissions), 39.7% (secondary particles) and 16.2% (resuspended dust). From 2015 to 2019, a dramatic decline was observed in the average percentage contributions of coal and biomass combustion, but the traffic emission source showed an increasing trend. For spatial variations, the high coefficient of variation (CV) values of coal and biomass combustion and industrial emissions indicated their higher spatial difference in Chengdu. High contributions of resuspended dust occurred at sites with intensive construction activities, such as subway and airport construction. Combining the PMF results, we developed the source weighted potential source contribution function (SWPSCF) method for source localization. This new method highlighted the influences of spatial distribution for source contributions, and the effectiveness of the SWPSCF method was well-evaluated.

## 1 Introduction

PM$_{2.5}$, fine particulate matter with aerodynamic diameter less than 2.5 μm, is a complex heterogeneous mixture of chemical constituents originating from a variety of sources (Bressi et al., 2013; He et al., 2019; Kelly and Fussell, 2012). Numerous epidemiological studies have reported associations between PM$_{2.5}$ and adverse human health effects (Bell Michelle et al., 2007; Yang et al., 2018b; Ostro et al., 2010; Philip et al., 2014), and have attracted broad attention to PM$_{2.5}$ in public, in the past decades. The link between urbanization and the spatiotemporal variability of PM$_{2.5}$ has been studied (Zhang et al., 2015; Li et

al., 2016). PM$_{2.5}$ generally presents an increasing trend with urbanization (Yang et al., 2018a). In addition, multiple policies were conducted by governments to alleviate the pollution (Yan et al., 2018; Cai et al., 2017). The urbanization stage and emphasis on pollution prevention policies vary greatly in both time and space (Wang et al., 2018a; Gurjar et al., 2016; Seto et al., 2017), causing significant spatiotemporal heterogeneity in the distribution of PM$_{2.5}$. Thus, a thorough understanding of the spatiotemporal and policy-related variations of PM$_{2.5}$ is necessary to investigate the relationship between urbanization and

PM$_{2.5}$. The spatiotemporal variability of PM$_{2.5}$ with the impact of urbanization has been reported in previous studies (Li et al., 2016; Timmermans et al., 2017; Zhang et al., 2019; Yang et al., 2020; Seto et al., 2017), among which a small number of publications focused on the analysis of PM$_{2.5}$ composition and sources (Lin et al., 2014; Yan et al., 2018). However, there is a lack of research on multiple sites and long-term sampling of PM$_{2.5}$ composition over a city-sized area (Dai et al., 2020; Xu et al., 2020a; Fang et al., 2020). Systematic measurements based on multiple sites and long-term observations can provide

valuable data for a comprehensive understanding of PM$_{2.5}$ characteristics and variations. Related studies are critical for promulgating targeted control policies from the perspective of urbanization.

In a city-sized area, there exist a large number of natural and anthropogenic sources of particulate matter, such as soil or road dust, vehicle exhaust, biomass combustion, sea salt, and smoke from forest fires, all of which show large spatiotemporal

variations (Zhang et al., 2015; Zhang et al., 2013; Mirowsky et al., 2013; Yang et al., 2018b). It is essential to identify and apportion PM$_{2.5}$ sources to provide targeted control policies. To date, receptor models have been applied in a number of source apportionment studies of PM$_{2.5}$, including factor analysis models (such as PCA-MLR, PMF, UNMIX, and ME2) and chemical mass balance (CMB) techniques (Shi et al., 2009; Choi et al., 2015; Hasheminassab et al., 2014; Liu et al., 2015). These receptor models have proven to be effective methods for identifying and apportioning sources. Furthermore, to identify the

likely source regions for a receptor site, a number of trajectory statistical methods have been widely applied, including concentration field (CF), concentration weighted trajectory (CWT), and potential source contribution function (PSCF) (Chen et al., 2011; Gebhart et al., 2011; Riuttanen et al., 2013; Kulshrestha et al., 2009b). In the traditional PSCF method, the source localization is mainly based on the number of trajectory endpoints that fall in the targeted grid cell. However, it should not be ignored that the sources showed discrepant spatial distribution patterns over the studied region. When trajectories pass over

the grid cell in which a source category shows high local contributions, the probability of potential contribution for this grid cell should theoretically be relatively high. Accordingly, we developed a source weighted PSCF (SWPSCF) method that combines PMF with PSCF and considers the spatial distribution of contributions for each source category. The SWPSCF can be employed as a valuable tool to obtain more precise estimates of potential source areas.

In China, megacities have experienced frequent air pollution events in response to rapid economic growth and urbanization (Li et al., 2016; Luo et al., 2018), which has prompted governments to take various measures to improve air quality. Chengdu, typical megacity in China, can represent an illustrative example of urbanization in a metropolitan region. Since the implementation of pollution prevention policies, namely the Air Pollution Prevention and Control Action Plan (APPCAP), Blue Sky Protection Campaign, and the thirteenth Five-Year Plan (Cai et al., 2017), air quality in Chengdu has been remarkedly

improved; thus, Chengdu serves as a useful case study in which we can investigate the spatiotemporal and policy-related variations of PM$_{2.5}$. In this study, we investigated the spatiotemporal and policy-related variations of PM$_{2.5}$ composition and sources in Chengdu at multiple sites based on a long-term observation. The positive matrix factorization (PMF) model was applied to estimate PM$_{2.5}$ source contributions. The SWPSCF method was then applied to identify the potential source locations. The main objectives of this study were: (i) to analyse the long-term spatiotemporal variations of PM$_{2.5}$ composition among

multiple zones in different urbanization levels; (ii) to determine PM$_{2.5}$ sources and their contributions, and to evaluate the effectiveness of the SWPSCF method in potential source localization; and (iii) to explore the spatiotemporal evolution of sources along with changes in urbanization and related policy orientation. We propose the findings of this research will be

helpful for a comprehensive understanding of the impact of the urbanization process and control policy on variations in PM$_{2.5}$ composition and sources in different zones, which can provide basic information for future epidemiological studies. It is of vital importance for further formulating emission reduction policies in China and other developing and polluting countries.

## 2 Method and Materials

### 2.1 Sampling sites and sampling

We collected PM$_{2.5}$, from Chengdu (102° E to 104° E, 30° N to 31° N), a city in the southwest of China with a population of 16.33 million and an area of 14,605 km$^2$. As an important metropolitan region in western China, Chengdu is undergoing rapid urbanization and is attracting an increasing number of residents. At the same time, PM pollution has received much attention. To improve air quality, the Chengdu government adopted several measures, including limiting the driving area of highly polluting vehicles and setting specific hours for driving in, adjusting industrial structures, and implementing energy substitution. Considering the heterogeneous spatial distribution of population, economy, industry, and construction activities, the degree of urbanization as well as air quality varies greatly in different sections of the Chengdu, and the emphasis on corresponding policies also varies across the city. As shown in Fig. 1, sampling was conducted at 19 sites in Chengdu (detailed information is shown in Table S1). Based on the specific characteristics, 19 sampling sites were clustered in different zones for purpose of comparison. The following sites—being located in the core zone of Chengdu and having developed earlier in the urbanization process—have high population and high traffic levels: Environment Protection Building (QY1), Chengdu University of Technology (CH1), and botanical garden (JN1). Combining the city structure and urbanization levels, Chengdu residents are accustomed to defining regions surrounded by the third circle road as "layer 1", and the location of QY1, CH1 and JN1 are in accordance with the extent of layer 1. The following sampling sites are included in the outer circle of layer 1: Qingbaijiang (QBJ2), Xindu (XD2), Pidu (PD2), Wenjiang (WJ2), Shuangliu (SL2), Tianfu (TF2) and Longquanyi (LQY2). This outer circle developed later than the area corresponding to layer 1. Accordingly, these sites are grouped together as the second zone and correspond to layer 2. Among the sampling sites in layer 2, QBJ2, XD2, WJ2, and SL2 are characterized by intensive industrial factories, and TF2 has frequent construction activities. The coordinates of factories in some key industrial sectors are presented in Fig. S1. The remaining nine sites are located in the outer-most zone of Chengdu and correspond to layer 3: Jintang (JT3), Pengzhou (PZ3), Dujiangyan (DJY3), Chongzhou (CZ3), Dayi (DY3), Qionglai (QL3), Pujiang (PJ3), Xinjin (XJ3), and Jianyang (JY3). The urbanization level of layer 3 was lower than that of layers 1 and 2. In addition, because air pollution is usually heavy in winter, the sampling campaign was conducted in winter from 2015 to 2019, lasting approximately 15 days each year. The detailed sampling periods for the sampling sites in 2015–2019 are listed in Table S2. Although several selected sampling sites may not be fully consistent each year, this small difference does not influence the reflection of spatiotemporal variations in Chengdu. A total of 836 PM$_{2.5}$ samples were collected from 19 sites for analysis.

The sampling campaign was simultaneously conducted using two medium-volume air samplers (TH-150C; Wuhan Tianhong Ltd., China) with an airflow rate of 100 L min$^{-1}$ at each site. One sampler placed quartz filters to collect PM$_{2.5}$ for analysing organic carbon (OC), elemental carbon (EC) and ions. The other sampler placed polypropylene filters to analyse elements in PM$_{2.5}$. Samples were collected daily for 22 h (from approximately 11:00 am to 09:00 am local time, GMT+8) at 19 sites. Average temperature (℃), cumulative volume (L), and standard volume (L), were recorded. When it rained, we stopped the sampling campaign. The air flow rate was corrected by a flowmeter before each sampling period. Collected samples were stored in a layer of aluminium foil in a freezer at -20 ℃ until weighing and analysis. The mass of PM$_{2.5}$ was determined by the difference in weight of the filter before and after sampling. Before sampling, blank quartz filters and blank polypropylene filters were baked at 600 ℃ for 4h and 60 ℃ for 3 h, respectively. For the process of weighing, filters were weighed at a temperature of 20±1 ℃ and humidity of 40±5% for 48 h. The weights of the filters can be obtained using a microbalance

(METTLER TOLEDO UMX) with a sensitivity level of 0.01 mg. Each filter was weighed twice, and the final weight was equal to the average of the two values (the difference was less than 0.05 mg).

## 2.2 Chemical analysis and quality assurance/ quality control (QA/QC)

The OC, EC, ions and elements were detected using the thermal/optical carbon aerosol analyser (DRI model 2001A; Desert Research Institute, USA), an ion chromatograph system (ICS-900; DIONEX, USA), and Inductively Coupled Plasma Atomic Emission Spectrometer (ICAP 7400 ICP-AES; Thermo Fisher Scientific, USA), respectively. The following is a brief description of the pre-treatment procedure, chemical analysis and QA/QC; more detailed information is provided in our previous works (Tian et al., 2016; Tian et al., 2014; Bi et al., 2007; Kong et al., 2010; Xue et al., 2010).

### 2.2.1 Organic carbon (OC) and elemental carbon (EC) analysis

OC and EC were analysed based on a hole with a quartz filter of $0.588$ cm$^2$. The thermal/optical carbon aerosol analyser detected OC1, OC2, OC3, and OC4 in a pure helium atmosphere at temperatures of 140, 280, 480, and 580 °C, respectively. Similarly, the oven temperature was increased to 540 °C, 780 °C, and 840 °C for EC1, EC2, and EC3 analyses, respectively, in a 2% $O_2$ atmosphere. Organic pyrolyzed carbon (OPC) was also detected after adding oxygen. Finally, the OC and EC concentrations were calculated using Eqs. (1) and (2), respectively. QA/QC was conducted using a calibration process. The method detection limit was 0.82 μg C cm$^{-2}$ for OC, and 0.20 μg C cm$^{-2}$ for EC.

$$OC = OC_1 + OC_2 + OC_3 + OC_4 + OPC \tag{1}$$
$$EC = EC_1 + EC_2 + EC_3 - OPC \tag{2}$$

For QA/QC, a system stability test (three-peak detection) is required before and after detecting samples and the relative standard deviation should not exceed 5%. The sample was reanalysed for every ten samples.

### 2.2.2 Ions analysis

Ions including Cl$^-$, SO$_4^{2-}$, NO$_3^-$, and NH$_4^+$ were measured on a one-eighth sample. The portion was cut into small pieces directly into tubes and ultrasonically extracted with 8 mL deionized water for 20 min. The tubes used during extraction were cleaned three times using an ultrasonic cleaner. After extraction, the solution was stored in a refrigerator for 24 h. The supernatant was sucked by needle tubing and injected into a vial through two 0.22 μm filters. The obtained solution was analysed using ion chromatography to determine the ions. Anions were analysed using Dionex IonPac CS12A (4 mm) analytical column equipped with Dionex IonPac CG12A (4 mm) guard column, Dionex CSRS-500 (4 mm) was used as the suppressor, and methane sulfonic acid (20 mL of 99% methane sulfonic acid solution diluted to 2000 mL) was applied as the eluent. Cation analysis was conducted using Dionex IonPac AS22 (4 mm) as the analytical column, Dionex IonPac AG22 (4 mm) as the guard column, Dionex ASR-500 (4 mm) as the suppressor, NaHCO$_3$ (0.14 mol L$^{-1}$) and Na$_2$CO$_3$ (0.45 mol L$^{-1}$) as the eluent. A conductivity detector was equipped for both anion and cation analysis with an injection volume of 0.5-0.8 mL and an eluent flow rate of 1.2 mL min$^{-1}$.

For QA/QC, the RSD was calculated more than three times to hold the value at a lower value (<5%). A standard sample test was performed using certified reference materials (CRM, produced by National Research Center for Certified Reference Materials, China) to ensure QA/QC. The spiked recoveries of ions ranged from 96.0% to 110.0%, as reported in Table S3.

### 2.2.3 Elements analysis

The microwave acid digestion method was applied to detect the following elements: Al, Fe, Mg, Ca, Na, K, V, Cd, Pb, Si, Zn, Cu, Cr, As, Ni, Co, Mn, and Ti. A 10 mL mixed digestion solution (2 mL HNO$_3$, 6 ml HCl, and 2 ml H$_2$O$_2$) was added to

digest one-eighth sample pieces, and the digestion process was conducted by a four-stage microwave digestion procedure of the microwave-accelerated reaction system (MARS; CEM Corporation, USA): the temperature was increased to 120 °C in 10 min, held for 8 min, reached 150 °C in 3 min, held for 8 min, reach 180 °C in 3 min, held for 8 min, and then reached 200 °C in 3 min and held for 10 min. Subsequently, the digestion solution was transferred into a PET bottle, and the solution was diluted to 25 mL with deionized water for further analysis using an inductively coupled plasma atomic emission spectrometer. During analysis, the target element can radiate the characteristic spectral lines, the intensity of which is directly proportional to the concentration of the element.

For QA/QC, a single-point calibration and blank tests were conducted for every ten samples. Single-element standards purchased from CRM were used for the calibration of each element. The determined RSD was below 10%, and the spiked recoveries for all elements varied between 85.5% and 113.1%, as listed in Table S3.

### 2.3 Positive Matrix Factorization (PMF)

The PMF model is a widely used bilinear receptor model. The goal of this model is to identify and quantify the source contribution of contaminants by solving the following equation (Eq. (3)):

$$x_{ij} = \sum_{k=1}^{p} g_{ik} f_{kj} + e_{ij} \tag{3}$$

where i, j, and p are the number of samples, chemical species, and factors, respectively, $x_{ij}$ is the concentration of the $j_{th}$ species in the $i_{th}$ sample, $g_{ik}$ is the contribution of the $k_{th}$ source to the $i_{th}$ sample, $f_{kj}$ is the concentration of the $j_{th}$ species from the $k_{th}$ source, and $e_{ij}$ is the residual for each sample/species (Paatero, 1997; Paatero and Tapper, 1994).

We input the measured speciated data as the matrix X of i by j dimensions; then, the PMF model can divide it into two matrices: factor contributions (G) and factor profiles (F). The non-negativity constraint was also introduced to ensure a positive value for each source contribution. In the decomposition process, the model is run several times by applying the least-squares method to minimise the objective function Q (Eq. (4)), and in this case, the obtained solutions of G and F are considered optimal:

$$Q = \sum_{i=1}^{n} \sum_{j=1}^{m} \left( \frac{e_{ij}}{u_{ij}} \right)^2 \tag{4}$$

where $u_{ij}$ is the uncertainty of the $j_{th}$ chemical species in the $i_{th}$ sample. The model required both concentration data and the uncertainty of the species in each sample. The equation-based uncertainty is calculated as follows (Eq. (5)):

$$u_{ij} = \begin{cases} \frac{5}{6} \times MDL, & c_{ij} \leq MDL \\ \sqrt{\left( Error\ Fraction \times c_{ij} \right)^2 + (0.5 \times MDL)^2}, & c_{ij} > MDL \end{cases} \tag{5}$$

where $c_{ij}$ is the concentration of chemical species in each sample, and MDL is the method detection limit for each component.

In this study, the EPA PMF 5.0 was applied for the source apportionment of $PM_{2.5}$. Because many concentrations of Cr and Co were under the MDL, they were not input for the apportionment, and a total of 22 chemical species of 836 samples at 19 sites from 2015 to 2019 were simulated. The detailed source apportionment results are reported in Section 3.3, and more information on the PMF model is described in the PMF 5.0 User Guide.

### 2.4 Source Weighted Potential Source Contribution Function (SWPSCF)

The PSCF model is a conditional probability that was applied to identify the source regions of PM2.5 masses to the receptor site. In this study, the backward trajectories were modelled using the MeteoInfo, a GIS application which enables the user to visualize and analyse the spatial and meteorological data with multiple data formats, which is available at http://www.meteothinker.com/. The required meteorological data were obtained from the National Centers for Environmental

Prediction (NCEP) global reanalysis data, which are available from the National Oceanic and Atmospheric Administration (NOAA's) Air Resources Laboratory (ARL) in a format suitable for transport and dispersion calculations. A detailed dataset can be obtained from the NOAA ARL FTP server (https://ready.arl.noaa.gov/archives.php). Using MeteoInfo modelling, 12 h backward trajectories starting from the receptor site at 500 m above ground level were generated with 6 h time intervals during all sampling periods. The 24 h and 72 h backward trajectories were also simulated in the process of parameter selection. The results suggested that regions passed over by 24 h and 72 h backward trajectories were far more widespread than those in our study area. The 12 h backward trajectories covered the most suitable range. In addition, it is possible to apply 24 h and 72 h trajectories when future studies refer to larger regions. In addition, the selection of the time interval showed little influence on the results.

The PSCF model divided the region where trajectories passed over into 0.1°×0.1° grid cells and computed the PSCF values of all grid cells in the domain. For the receptor site, the daily concentrations were assigned to the grid cells along related trajectories, and a certain threshold criterion value was selected. When the concentration in one grid cell was above the threshold value, there exists a probability that sources located in this grid cell have an influence on the receptor $PM_{2.5}$. A higher PSCF value indicates higher probability. The PSCF values are defined by Eq. (6) (Han et al., 2007):

$$PSCF_{ij} = (\frac{m_{ij}}{n_{ij}})W_{ij} \tag{6}$$

where $n_{ij}$ is the total number of trajectory endpoints that fall into the grid cell (i, j), and $m_{ij}$ is the number of trajectory endpoints when their corresponding contributions exceed the criteria value. $W_{ij}$ is a weight function (Eq. (7)) used to reduce uncertainty when specific grid cells have a small number of trajectory endpoints (Polissar et al., 2001; Lee and Hopke, 2006):

$$W_{ij} = \begin{cases} 1.0 & 3n_{ave} < n_{ij} \\ 0.7 & 1.5n_{ave} < n_{ij} < 3n_{ave} \\ 0.4 & n_{ave} < n_{ij} < 1.5n_{ave} \\ 0.2 & n_{ij} < n_{ave} \end{cases} \tag{7}$$

where $n_{ave}$ is the average number of endpoints in each grid cell.

When trajectories passed over a grid cell in which a certain source category showed a high local contribution, the probability of the potential contribution of this grid cell should be relatively high. Thus, we introduced another weighted function $SW_{ij}$ that represents the ratio of the source contribution in grid cell (i, j) to the average contribution in the whole study area. The $SW_{ij}$ is calculated using Eq. (8). The SWPSCF value is expressed in Eq. (9).

$$SW_{ij} = c_{ij}/c_{ave} \tag{8}$$

$$SWPSCF = SW_{ij} \times PSCF \tag{9}$$

where $c_{ij}$ is the source contribution of each source category in the grid cell (i, j), and is available using the Kriging interpolation algorithm; $c_{ave}$ is the average source contribution of this source category of all sampling sites in the entire study area.

**2.5 Hierarchical cluster analysis (HCA)**

The similarity analysis of $PM_{2.5}$ composition among the 19 sampling sites from 2015 to 2019 was conducted using hierarchical cluster analysis. Cluster analysis, a technique used to identify groups that have similar characteristics, can be broadly classified as hierarchical and non-hierarchical (Govender and Sivakumar, 2020; Saxena et al., 2017). By recursively finding nested clusters, hierarchical clustering repeatedly combines the two closest groups into one larger group (Xu et al., 2020b), and finally generates a dendrogram. The algorithm is implemented mainly by the following steps (Govender and Sivakumar, 2020):
Step 1: Determine each observation as the initial cluster.
Step 2: Measure the distance between clusters for quantifying the similarity between objects.

Step 3: The closest pairs of clusters are merged into a single cluster, and the distance matrix is recalculated.

Step 4: Repeat steps 2 and 3 until all observations are integrated into a single cluster.

To guarantee the effectiveness of the algorithm, appropriate methods should be selected according to the properties of specific objects. An introduction on the specific choice of the distance metric and linkage function is added in the Supplement. In this study, the HCA was conducted using IBM SPSS Statistics 25, and the results were confirmed to be similar by using different

distance metrics and linkage methods. Based on the comprehensive consideration, the HCA based on the cosine distance and average linkage method was selected. By cutting the dendrogram at an appropriate distance, $PM_{2.5}$ samples that have similarities in chemical species can be grouped into the same cluster.

## 3 Results and Discussion

### 3.1 Spatiotemporal variations of $PM_{2.5}$ concentrations

The spatiotemporal variations in $PM_{2.5}$ concentrations for layers and sites in 2015–2019 are depicted in Fig. 2. The detailed $PM_{2.5}$ concentrations are summarised in Table S4. Due to the slight difference in the selected sampling sites in layers 2 and 3 in each year, both layers and sites were discussed for a better understanding of the $PM_{2.5}$ variability. For spatial distribution, the average $PM_{2.5}$ concentrations over five years were 126 $\mu g \cdot m^{-3}$, 133 $\mu g \cdot m^{-3}$, and 121 $\mu g \cdot m^{-3}$ for layers 1, 2, and 3, respectively. Layer 1, the most urbanized area in Chengdu, suffered severe traffic pollution; however, stricter control policies

were implemented by local governments in this area. The high $PM_{2.5}$ concentration in layer 2 may be caused by strong industrial activities and extensive construction activities at QBJ2, XD2, WJ2, SL2, and TF2. Layer 3 was characterized by the lowest urbanization level in Chengdu, although weak emissions of old chemical industries and small coal-fired boilers were observed at XJ3, PZ3, CZ3, and DY3; there were fewer vehicles than layer 1 and fewer factories than layer 2, explaining the relatively low levels of $PM_{2.5}$ in the area.


$PM_{2.5}$ concentrations in three layers showed similar temporal variation, which averagely declined from 174 $\mu g\ m^{-3}$ in 2015 to 95 $\mu g\ m^{-3}$ in 2019, except for a small increase in 2017 (134 $\mu g\ m^{-3}$), indicating the effective control measures in Chengdu in recent years. Fig. S2 shows the temporal variation of daily $PM_{2.5}$ concentrations and annual average $PM_{2.5}$ concentrations for each site. The large number of sampling data from all filters further demonstrates the temporal changes in $PM_{2.5}$ concentrations

over time, as described above. The results of the statistical analysis, using the two tailed matched t-tests for $PM_{2.5}$ concentrations at sampling sites between 2015 and 2019, are summarized in Table S5. As seen in the table, there was a significant decreasing trend in the level of $PM_{2.5}$ in the period 2015-2019. A more obvious decline was observed at the sites in layer 1. In 2015, the $PM_{2.5}$ concentration was the highest in layer 1; however, since 2016, the highest $PM_{2.5}$ level has been transferred from layer 1 to layer 2. This may be influenced by the fact that the coal-burning ban was promulgated the earliest

in layer 1. The government published Chengdu's Air Pollution Prevention and Control Regulation in each year and introduced a number of specific measures, including the substitution of clean energy boilers for existing coal-fired boilers, which was accelerated in 2016 in layer 1. $PM_{2.5}$ concentrations at several sites in layer 2 exhibited a minor elevation: for example, $PM_{2.5}$ levels at WJ2 and SL2 increased in 2018. This may be associated with construction and industrial activities in this region. Temporal variations of sites in layer 3 are not discussed due to the deficiency of $PM_{2.5}$ concentrations in many studied years.

**3.2 Spatiotemporal variations of chemical composition**

Research on the chemical composition of $PM_{2.5}$ can be helpful in identifying the source changes and the effectiveness of related policies. In Fig. 3 we present the fractions of the main chemical species (%) in $PM_{2.5}$ at each site during the winter in the period 2015–2019, reflecting the relative importance of species under different $PM_{2.5}$. The average fractions of $PM_{2.5}$, in the order of

$OC > NO_3^- > SO_4^{2-} >$ crustal elements (the sum of Al, Si, Ca, Ti, and Fe) $> NH_4^+ > EC > Cl^-$, constituting 17.2%, 13.5%, 11.0%, 8.3%, 5.7%, 5.4%, and 2.3% of the $PM_{2.5}$ mass, respectively.

To identify the similarity and diversity of $PM_{2.5}$ composition among the sampling sites and years, hierarchical cluster analysis (HCA) (based on cosine distances) of chemical composition (%) was carried out at each sampling site for five years (2015-2019). The results are shown in Fig 4. Four clusters were identified, and the results showed a strong correlation with years: Cluster 1 (C1) consisted of most sites in 2018 and 2019; sites in 2016 and 2017 were classified as cluster 2 (C2); cluster 3 (C3) included all the sites in 2015; and 2016DJY3, the only site far from the other sites, was separated as cluster 4 (C4). A total of thirteen samples were collected at 2016DJY3, and both the sampling number and duration were similar to samples collected at other sites in 2016. As a typical background site in Chengdu, DJY3 is surrounded by plants and agricultural activities, so it is featured by the distinctive $PM_{2.5}$ compositions with markedly high $NH_4^+$ and crustal elements. This explains the particular HCA result of the C4 well. The meteorological data ([https://rp5.ru/](https://rp5.ru/)) during the sampling period from 2015 to 2019 are shown in Table S6, reflecting the similar meteorological conditions in the studied years, which highlighted the importance of the source variations for the clustering results. There was a special case where the sites of layer 3 in 2019 belonged to C2 rather than C1, indicating that the $PM_{2.5}$ composition for layer 3 in 2019 was more similar to that for other layers two or three years ago. This can be explained by the fact that urbanization levels varied between the layers in Chengdu. As the outer-most zone of Chengdu, layer 3 lagged behind layer 1 and layer 2 in the urbanization, which contributed to the similar characteristics in air quality between current layer 3 and previous other layers. The HCA results indicated an incredible need to investigate the variations of $PM_{2.5}$ composition in both time and space.

### 3.2.1 Spatial variations of chemical composition

To investigate the spatial similarities and differences of chemical composition, the HCA was also applied based on the chemical composition (%) at sampling sites for each year, and the cluster results and their averaged species fractions are listed in Fig. S3.

The chemical composition of the clusters in 2015-2019 is shown in Fig. 5. Taking as an example the first cluster in 2015, we defined it as 2015C1. Spatial differences were observed each year. The clusters 2015C4, 2016C4, 2017C1, 2018C2, and 2019C1 always showed higher OC fractions: 20.9%, 14.6%, 20.5%, 17.5%, and 23.3% of $PM_{2.5}$ mass, respectively. The higher OC fractions of these clusters were considered to occur at the contained sites, such as PZ3, JT3, CZ3, XJ3, JY3, and PJ3, and could be either directly emitted (primary organic carbon, POC) or indirectly formed in the atmospheric (secondary organic carbon, SOC) (Kanakidou et al., 2005; Zhong et al., 2021). The high POC was largely associated with the stronger fuel combustion and biomass burning. One possible reason is that there were more residential combustions (such as bulk coal and biofuel combustion) and small boilers with low combustion efficiency at PZ3 and XJ3; therefore, control measures for fuel combustion still need to be strengthened. During the sampling period, activities such as the burning of firewood by residents to produce smoked meat can contribute greatly to the OC level from biomass burning. Additionally, the formation of SOC was also responsible for the high OC level. SOC is generated from the oxidation of volatile organic compounds (VOCs) through homogeneous or heterogeneous reactions (Jang et al., 2002). VOC precursors come from both anthropogenic sources and plant emissions (Ait-Helal et al., 2014; Kleindienst et al., 2009). Previous studies (Zhao et al., 2018; Han et al., 2013; Yin et al., 2015) have reported high VOC emissions from industrial processes at PZ3, JT3, and other sites. Coal combustion in industries and thermal power plants were the main sources of industrial processes at PZ3 and JT3, respectively. Biogenic VOC emissions often occur at several agriculture sites such as JY3 and PJ3 because of the high vegetation coverage in these areas. High $NO_3^-$ levels in Chengdu were observed at PZ3 in 2015, and QY1 and CH1 in 2019. The high $NO_3^-$ levels at PZ3 in 2015 may be associated with the petrochemical industry. In 2019, the $NO_3^-$ level at PZ3 was lower than that in 2015, which might have been

influenced by the renovation of de-nitrification of the key industries. On the other hand, vehicle ownership in Chengdu markedly increased, especially in layer 1. Characterized by the most intensive vehicles, QY1 and CH1 experienced heavy traffic pollution. Crustal elements accounted for the highest proportion in layer 1 related clusters (2016C3, 2017C3, and 2018C4) with 10.5%, 9.9% and 8.3%, respectively. The subway construction in layer 1 of Chengdu can explain this result.

### 3.2.2 Temporal variations of chemical composition

With respect to the temporal variations of composition shown in Fig. 3, the fractions of OC and EC generally showed a decreasing trend from 2015 to 2018 and slightly increased in 2019 at most sites. The average fractions of OC were 19.1% and 15.5% in 2015 and 2018, respectively. EC accounted for 15.5% and 5.0% of $PM_{2.5}$, in 2015 and 2018, respectively. The OC and EC mainly come from the combustion of fuels, such as coal, gasoline, diesel and biomass (Wang et al., 2020). In Chengdu, coal is one of the important fuels for the industry, but has been strongly reduced by the government in recent years. Gasoline and diesel are mainly used in vehicles. The decrease in OC and EC fractions from 2015 to 2018 may be due to the decline in coal use for industries, which was consistent with the strict coal-burning ban in these years; however, as the vehicles became more important contributors, the OC and EC fractions increased in 2019. The absolute concentrations of $SO_4^{2-}$, $NO_3^-$ and $Cl^-$ are shown in Fig. S4. Publications have reported the use of $SO_4^{2-}$ and $Cl^-$ as coal-burning markers (Tian et al., 2014; Vassura et al., 2014). In the five years of the study, the average concentrations of both $SO_4^{2-}$ and $Cl^-$ sharply decreased, from 28 μg m$^{-3}$ to 8 μg m$^{-3}$ and from 6 μg m$^{-3}$ to 2 μg m$^{-3}$, respectively. The fractions of $SO_4^{2-}$ and $Cl^-$ also showed a decreasing trend, especially in 2016. However, the fractions of $NO_3^-$ showed a general increasing trend from 2015 to 2019. The average concentrations of $NO_3^-$ were found to decrease from 20 μg m$^{-3}$ in 2015 to 14 μg m$^{-3}$ in 2016, mainly resulting from the strongly promoted coal-burning ban policy, after that, $NO_3^-$ increased slightly to 16 μg m$^{-3}$ in 2019, which might be attributed to the gradually enhanced contribution of vehicles and use of natural gas. We also analyzed the $SO_4^{2-}/NO_3^-$ mass ratio, a qualitative indicator of sulfur versus nitrogen sources (Gao et al., 2015; Arimoto et al., 1996), and the summary is presented in Fig. S4 (d). Ratios at most sites exceeded 1 in 2015, dropped to less than 1 in 2016, and then declined steadily. Combined with the absolute concentrations of $SO_4^{2-}$ and $NO_3^-$ discussed above, the $SO_4^{2-}/NO_3^-$ mass ratio can also indicate decreasing coal combustion and increasing traffic emissions in Chengdu. This result is consistent with the slow reduction in $NO_X$ and the sharp decline in $SO_2$ emissions in China (Zhao et al., 2013; Wang et al., 2018b). For crustal elements, temporal variations were found to have close a relationship with the construction activities in Chengdu in 2015-2019.

### 3.3 Spatiotemporal variations of sources

PMF was used to quantify the source contributions in the studied areas, and five categories were selected with distinctively related source characteristics. Five sources were identified: traffic emissions, coal and biomass combustion, industrial emissions, secondary particles, and resuspended dust. The estimated source profiles in the form of species concentrations (μg·m$^{-3}$) and percentages of species sum (%) are shown in Fig. 6. Factor 1 contributed 15.5% of $PM_{2.5}$ and had high fractions of EC (70.0% of total EC) and OC (51.8% of total OC), which can be identified as traffic emissions (Xu et al., 2016). The relatively high $NO_3^-$ further revealed Factor 1 as the source of traffic emissions. The moderate fractions of Al, Si, Cu, Ni, and As in this factor may be associated with traffic activities, including resuspension of road dust, tire and brake wear, and oil burning (Kulshrestha et al., 2009a; Almeida et al., 2005; Amato and Hopke, 2012). Factor 2 was determined to be a coal and biomass combustion source. Coal combustion generally plays an important role in the energy structure of China. Identified as markers of coal combustion source, OC, EC, Cd, and $SO_4^{2-}$ exhibited high loadings in factor 2, with fractions of 25.8%, 20.3%, 61.9%, and 26.7%, respectively (Tian et al., 2016). The presence of biomass burning was indicated by the high fraction of $K^+$ in this factor (Amil et al., 2016; Richard et al., 2011). Factor 2 accounted for 19.7% of the total $PM_{2.5}$ mass concentration. Factor 3, which accounted for 8.8% of $PM_{2.5}$, was considered as an industrial emission source because of its high loadings of Fe (73.8%), Cu (70.7%), Mn (60.5%), Ti (85.5%), Ni (61.5%) and Mg (50.2%). These species are frequently used as source

markers for industrial emissions, including building materials and metallurgical production (Contini et al., 2014; Jiang et al., 2014). Factor 4 was characterized by nearly 76.7%, 61.2%, and 55.9% of $NO_3^-$, $NH_4^+$, and $SO_4^{2-}$, respectively, and no other high species. According to previous studies, $NO_3^-$, $SO_4^{2-}$, and $NH_4^+$ are indicative of secondary reactions (Richard et al., 2011; Wu et al., 2021). Consequently, factor 4 represented the secondary particle source, contributing to 39.7% of $PM_{2.5}$. Factor 5 was identified as resuspended dust, accounting for 16.2% of $PM_{2.5}$. The top three fractions of species were Al (84.2%), Ca (79.5%), and Si (56.5%), which are typical indicators of resuspended dust (Pant and Harrison, 2012).

### 3.3.1 Spatial variations of source contributions

In Fig. 7, we show the source contributions at each site from 2015 to 2019 in order to investigate their spatial variations. The coefficient of variation (CV), which is defined as the standard deviation divided by the mean, was used to investigate the spatial differences of each source category. As shown in Table S7, the CV values in this study indicate that coal and biomass combustion and industrial emissions show stronger spatial variations. For coal and biomass combustion sources, the percentage contribution was higher at CZ3 of layer 3 and QBJ2 of layer 2 than at other sites. The high contributions of industry sources mainly occurred in layer 2, including QBJ2, WJ2, PD2, SL2, and XD2, with fractions from 8.9% to 12.9%. Among the sampling sites mentioned above, CZ3 was characterized by intensive coal-fired boilers. QBJ2 contains large-scale iron, steel, and chemical plants. WJ2, PD2, SL2, and XD2 are located in areas of intensified development, including large factories of glass, food, and furniture. Therefore, the spatial distributions of $PM_{2.5}$, from coal and biomass combustion and industrial emissions, were strongly associated with industrial manufacturing plants. Additionally, the contributions of traffic emissions were higher in layers 1 and 2, with the percentage contributions in 2015–2019 ranging from 13.9% to 16.3% in layer 1 and from 11.6% to 17.5% in layer 2. The secondary particles had a higher contribution in layer 3. The fractions of secondary particles at QY1 and LQY2 also presented relatively high values of 44.5% and 49.9%, respectively. The spatial distribution of resuspended dust varied with human activity. The contributions were relatively higher in layer 1 in 2015–2018, which resulted from the construction of the urban subway. At JY3, high contributions from resuspended dust were attributed to the fact that Chengdu Tianfu International Airport was under construction. Overall, the spatial distributions of source contributions were in accordance with the characteristics and urbanization level of sites, highlighting the importance of site-specific and urbanization research in pollutant emission control.

To better consider the spatial distribution of contributions for each source category, the SWPSCF method was applied to identify the source regions to the receptor site based on the source contribution weight. In this study, we selected QY1 as the receptor site and the average contribution of each source category at QY1 as the threshold value. Both SWPSCF and PSCF values were calculated for each source category in the winter from 2015 to 2019. Examples of traffic emissions and coal and biomass combustion in 2015 and 2019 are shown in Fig. 8, and the differences were found in the PSCF and SWPSCF results. For coal and biomass combustion source in 2015 (Fig. 8(a)), the potential source regions were observed to concentrate to CZ3 after source weighting, and the SWPSCF values around QBJ2 were higher than the PSCF values, reflecting the strengthened influence of coal and biomass combustion sources at CZ3 and QBJ2. For the traffic emission source in 2019 (Fig. 8(b)), the identified potential source regions moved toward layer 1 after source weighting, which was in agreement with the spatial distribution of traffic emission contributions. As described above, the potential source regions identified after the source weighting could better reflect the spatial variations of source contributions, suggesting the effectiveness of the SWPSCF method in this study.

### 3.3.2 Temporal variations of source contributions

The temporal variations of source contributions at each site are summarised in Fig. 7. The contributions of traffic emissions at most sites showed an increasing trend from 2015 to 2019, because the number of vehicles increased rapidly. The average

percentage contributions of traffic emissions of layers 1 and 2 were in the order of 13.3% (in 2015) < 13.4% (in 2016) < 14.8% (in 2017) < 15.8% (in 2018) < 17.1% (in 2019). Contributions in layer 3 were not calculated because of the difference in sites in the studied year, but the tendency was consistent with the conclusions of layers 1 and 2. An obvious decline in the contribution of coal and biomass combustion can be observed in the studied years, especially in 2016. The average percentages of layers 1 and 2 declined from 33.2% in 2015 to 15.5% in 2016, and finally to 11.5% in 2019. The results indicated that notable success has been achieved in the control of coal-related sources in recent years. Industrial emissions showed the highest percentages in 2016 at some sites and presented a downward trend. The percentage of source contributions of secondary particles at most sites increased steadily each year. The average fractions of layers 1 and 2 from 2015 to 2019 were 29.8%, 40.0%, 41.2%, 46.0%, and 44.0%, respectively. For resuspended dust, the fractions in 2015 and 2016 were generally higher than those in other years, especially for sites in layer 1, which experienced major subway construction activity in previous years. In 2017–2019, the source contributions of resuspended dust remained stable, and some slight fluctuations could be attributed to local construction activities.

The above analysis of temporal variations provides insights into the changes of source structures in Chengdu: pollution from traffic and secondary aerosols played a more important role; sources from coal and biomass combustion and industrial emissions were effectively controlled; and resuspended dust always occurred along with the urban construction. All of this information can offer useful references for the government to further promulgate effective policies for atmospheric pollution prevention and reduction in China and other developing and polluting countries.

**4 Conclusions**

We investigated the spatiotemporal and policy-related variations of PM$_{2.5}$ composition and sources at 19 sites in Chengdu, based on a long-term sampling campaign in winter from 2015 to 2019. Considering the specific characteristics among sites, the variations were discussed in three layers of different urbanization levels. The results showed distinct spatiotemporal distribution patterns for both PM$_{2.5}$ composition and sources, linked to the process of urbanization and corresponding policies in the studied region.

During the sampling period, temporal variations of averaged PM$_{2.5}$ concentrations at sites in layer 1 showed the most obvious decreasing trend, caused by comparably strict control measures conducted in layer 1. The fractions of OC and EC declined from 2015 to 2018 and slightly increased in 2019 at most sites. The SO$_4^{2-}$/NO$_3^-$ mass ratio at most sites dropped less than 1 since 2016 and showed a decreasing trend, indicating decreasing coal combustion and increasing traffic emissions in Chengdu. The average percentage contributions of coal and biomass combustion sources declined from 2015 to 2019, reflecting the notable success in the control of coal-related sources in Chengdu. For spatial variations, the composition of PM$_{2.5}$ for layer 3 in 2019, was found to be similar to that for layers two or three years earlier, and this result indicates the considerable impact of differences in urbanization on air quality. The high CV values of coal and biomass combustion and industrial emissions are representative of the stronger spatial distribution patterns in Chengdu, the high percentage contributions of which usually occurred at sites with large-scale industrial factories and coal-fired boilers. Frequent construction activities in developing areas can considerably increase the percentage contribution of resuspended dust. The SWPSCF results were found to be significantly different from the PSCF results. The changes in the identified potential source regions after source weighting were in agreement with the spatial distribution of each source contribution. This study presented a perspective on the relationship between PM$_{2.5}$ and urbanization. Sampling activities that were conducted based on a five-year measurement at 19 sites in different urbanization levels provided valuable data for researchers. The results can be useful for further policy formulation in most developing and polluted countries, and provide basic information for future epidemiological studies.

*Data availability.* The coordinates of factories in some key industrial sectors presented in Fig. S1 are available at
     http://lbs.amap.com/api/webservice/guide/api/search/ (last access June 2$^{th}$, 2021). The MeteoInfo is available at
     http://www.meteothinker.com/ (last access September 5$^{th}$, 2021). Required meteorological data during SWPSCF modelling
     can be obtained from National Oceanic and Atmospheric Administration (NOAA) website,
     https://ready.arl.noaa.gov/archives.php (last access August 20$^{th}$, 2021). The provided meteorological data (Table S6) during
the sampling period in Chengdu is available at https://rp5.ru/ (last access June 7$^{th}$, 2021).

*Acknowledgments.* This study is supported by the National Natural Science Foundation of China (41977181).

*Author contributions.* Xinyao Feng were responsible for the writing of the paper and performing the SWPSCF model; Yingze
Tian provided the scientific idea and performed PMF experiments; Danlin Song and Fengxia Huang provided sampling data
     for analysis; Yingze Tian, Qianqian Xue and Yinchang Feng contributed to the project coordination.

*Competing interests.* The authors declare that they have no conflict of interest.

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

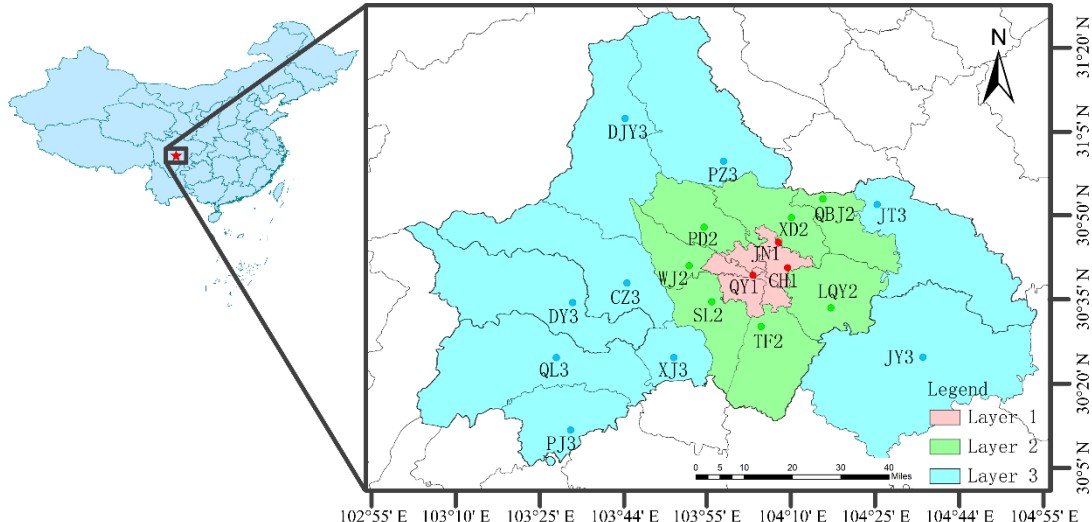

Figure 1: The locations of 19 sampling sites in Chengdu from 2015 to 2019.

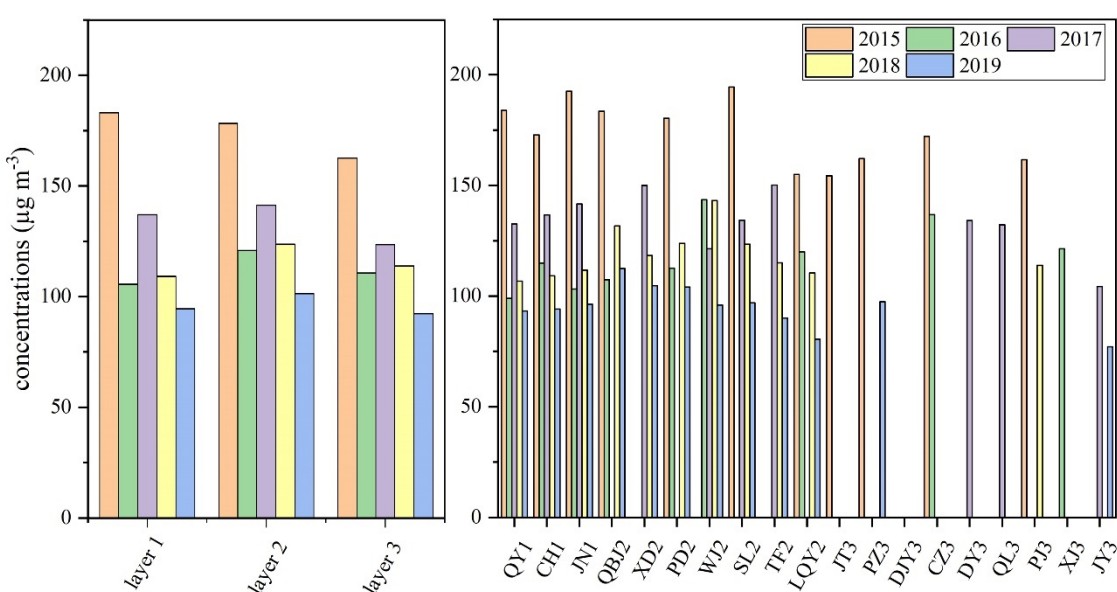

Figure 2. Spatiotemporal variations of PM$_{2.5}$ concentrations for layers and sampling sites in 2015-2019.

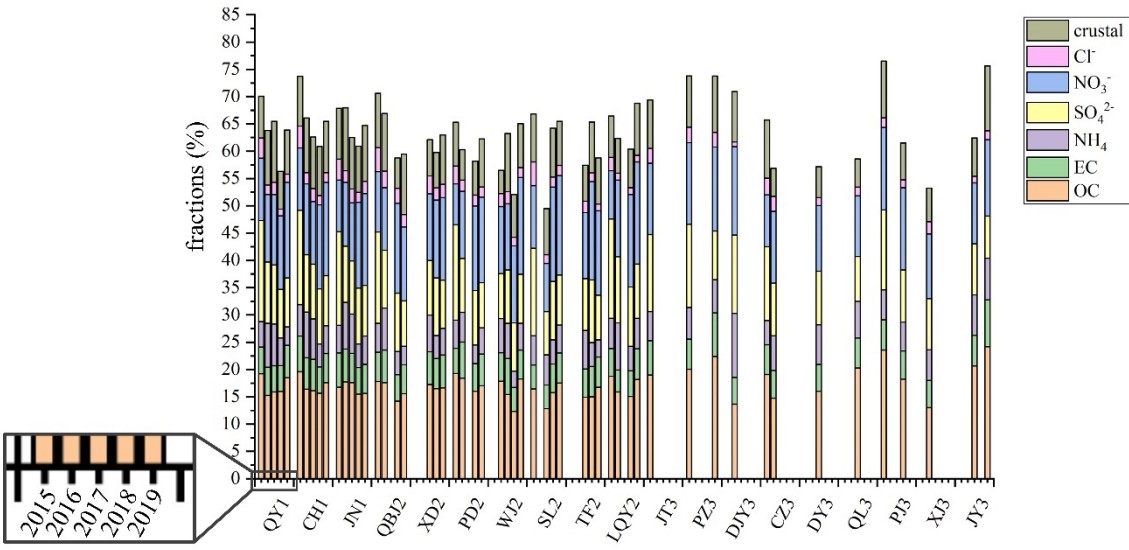

**Figure 3. The spatiotemporal variations of the fractions of main chemical species in PM₂.₅ at each site during winters in 2015 to 2019. Unit: %**

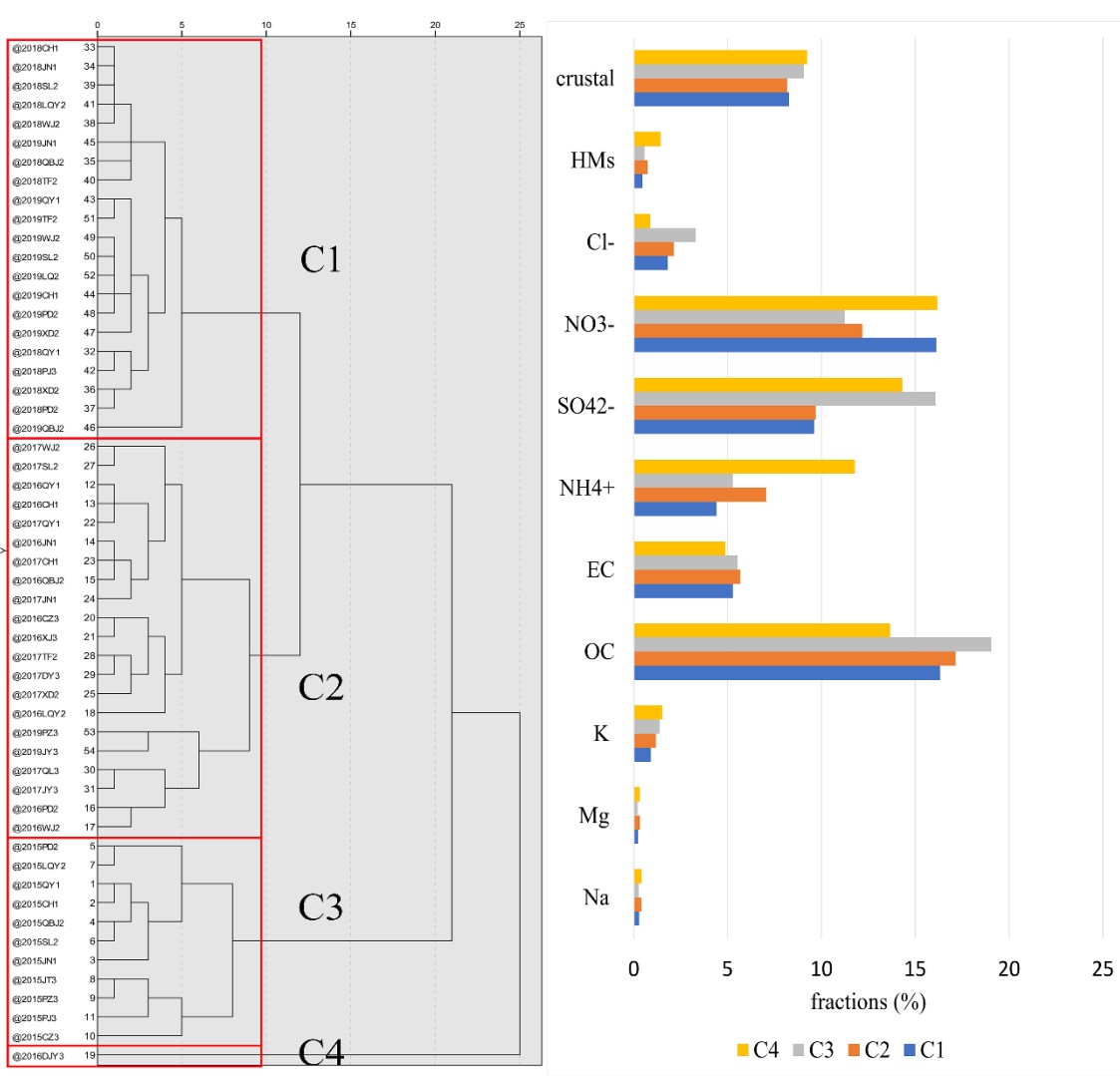


**Figure 4. The HCA results (based on cosine distances) of chemical species (%) at sampling sites for five years (2015-2019) and their averaged species fractions.**

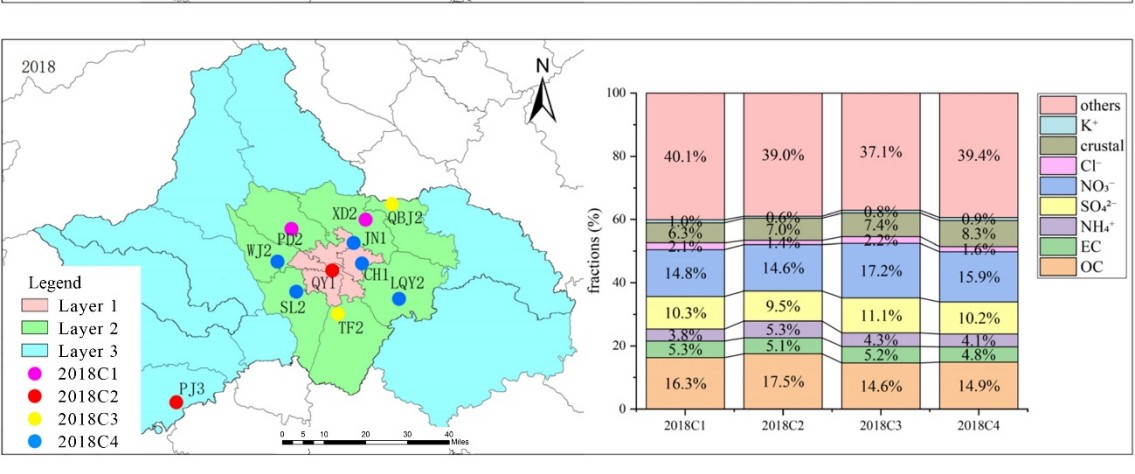


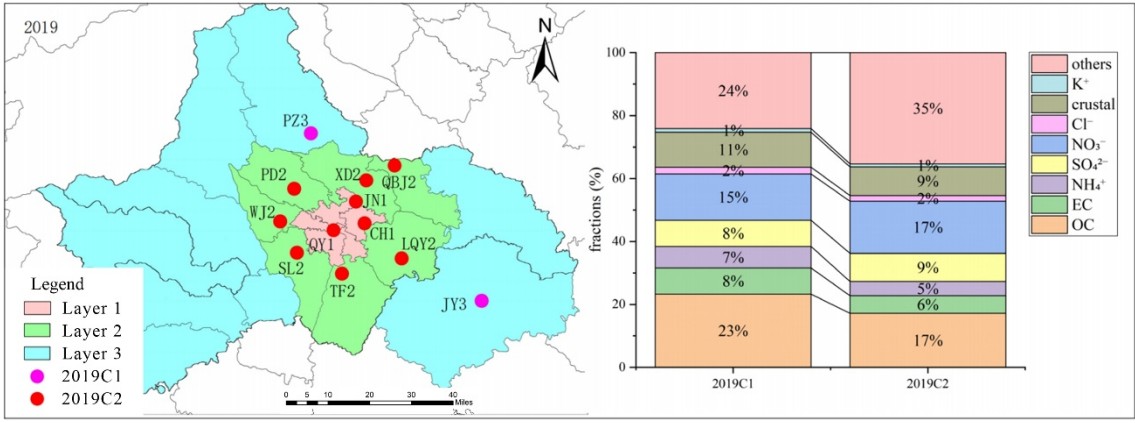

**Figure 5. Spatial distribution of PM₂.₅ compositions and fraction values of each cluster from 2015 to 2019. (i.e. 2015C1 refers to the first cluster of sampling sites in 2015).**

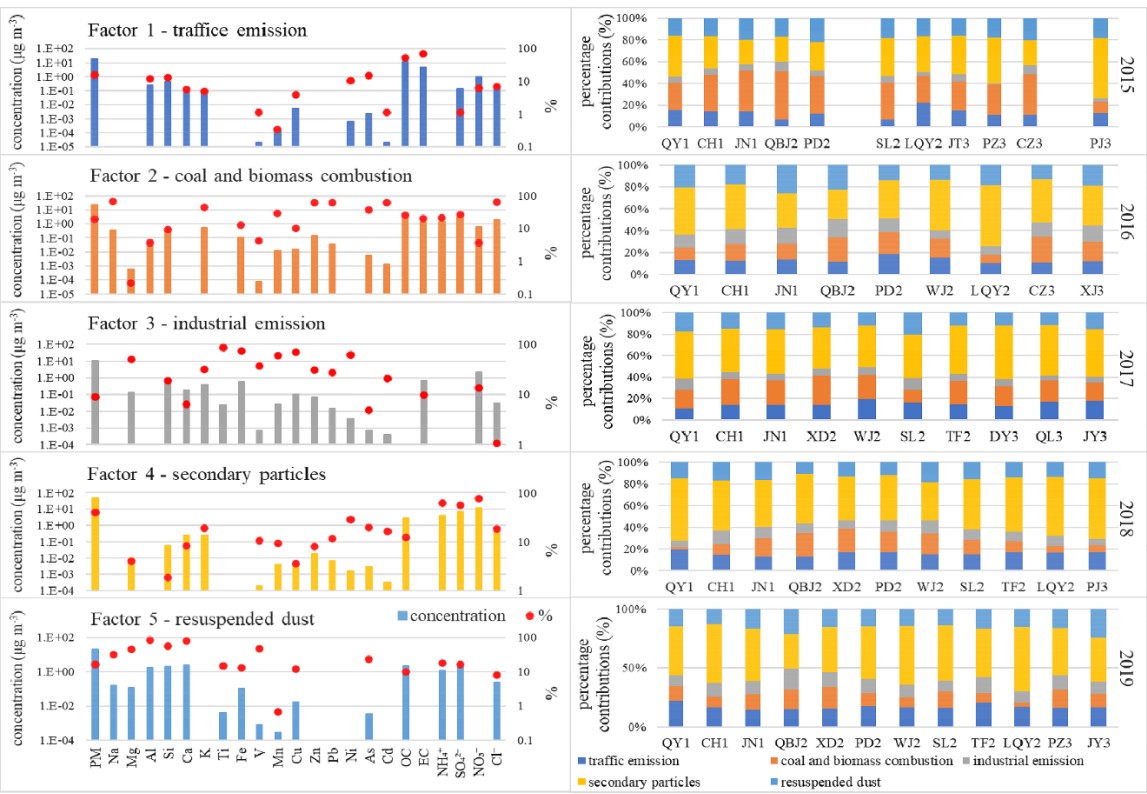

**Figure 6. Source profiles estimated by the PMF, in the form of species concentrations (µg m⁻³) and percentages of species sum (%).**

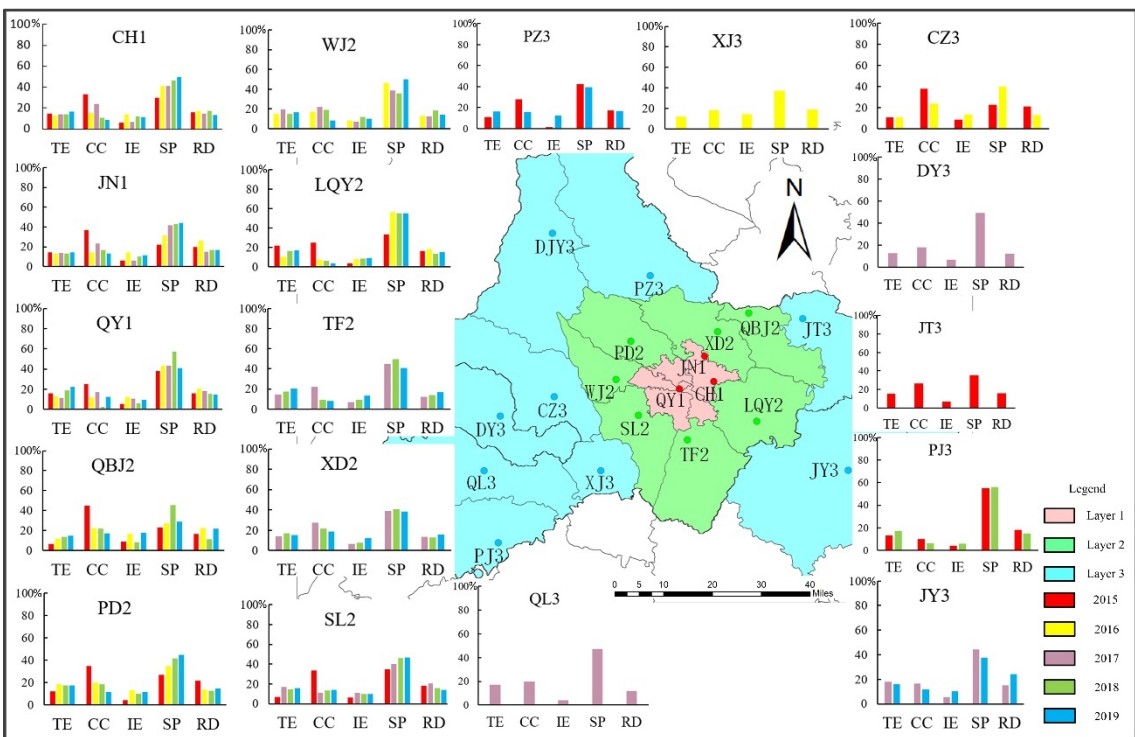

**Figure 7. Spatiotemporal variations of source contributions to total mass of PM$_{2.5}$ in Chengdu. (TE, CC, IE, SP and RD represent traffic emission, coal and biomass combustion, industrial emission, secondary particles and resuspended dust, respectively.)**

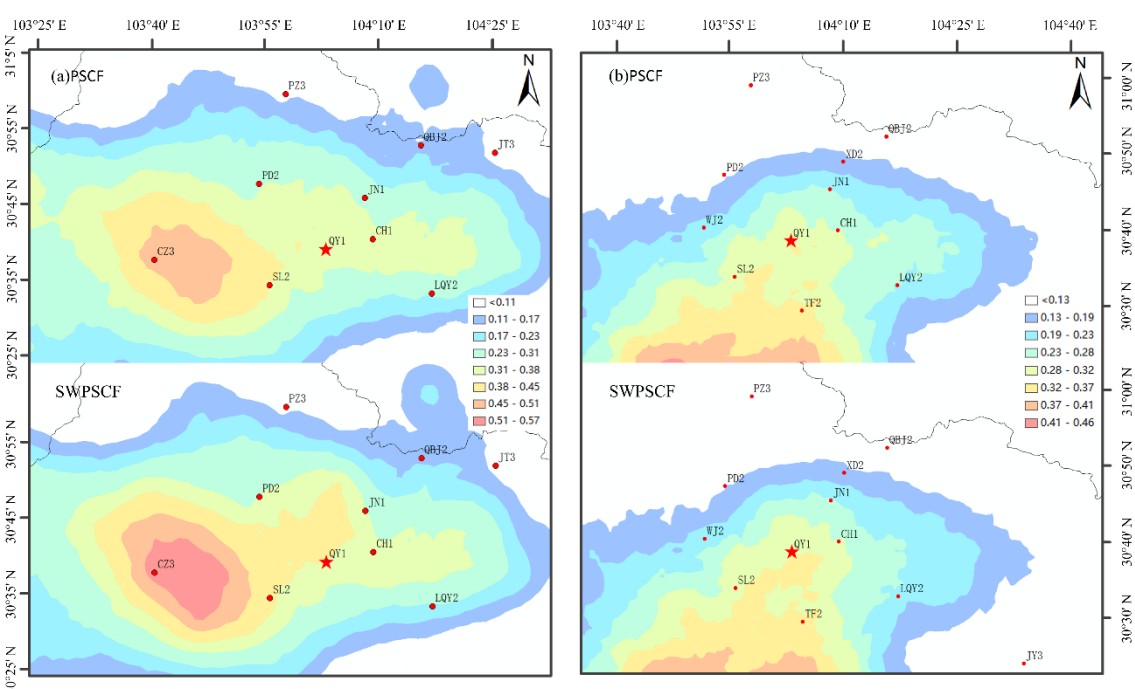

**Figure 8. Potential source locations identified by the PSCF and SWPSCF method: (a) coal and biomass combustion source in 2015;**
**(b) traffic emission source in 2019.**