# Peer review of "Measurement report: Spatiotemporal and policy-related variations of PM2.5 composition and sources during 2015-2019 at multiple sites in a Chinese megacity"

_Atmospheric Chemistry and Physics, 2021_

## Referee Comment (RC2)

Review Feng et al., ACPD 2021

"*Measurement report: Spatiotemporal and policy-related variations of PM2.5 compositions and sources during 2015-2019 at multisite of a Chinese megacity*"

Feng et al. present a measurement report on a PM2.5 filter data set that covers the years 2015-2019 at ~19 stations in the urban area of Chengdu, China. This is a valuable dataset, which displays the transformation of PM2.5 mass and composition in a rapidly developing urban area. Therefore, it is certainly within the scope of ACP. The general message is that regulations on coal combustion take effect within these five years, while PM2.5 due to traffic emissions and secondary particles increases. This is an important finding and worth publishing. However, in the current state, I cannot recommend the manuscript being published for several reasons: (1) the manuscript requires heavy language editing. Under "technical notes" I listed a few mistakes, I could have listed more, (2) the major message of the paper is that PM2.5 decreased between 2015 and 2019, but this is only based on the average of a large number of filter samples – no test for statistical significance (see major comment on chapter 3.1), (3) the methodology part is far from being complete – reproduction by fellow scientists would be impossible, (4) a few labels of clusters between figures and text are mixed up, (5) some of the data are interpreted in a highly speculative manner, often not convincingly supported by the data (see comment on chapter 3.2.1).
Overall, I conclude that the manuscript would need a major revision in order to meet the standards of ACP.

**Major comments:**

l. 22: A mass ratio is not necessarily an indicator for emission reduction. What is with the absolute values of sulfate and nitrate?

l. 25: A large fraction of secondary particles → this cannot be attributed to individual sources, such as traffic, coal combustion, industrial emissions? E.g. secondary sulfate can partly originate from coal combustion.

l. 28: What do you mean by "stronger distribution patterns."?

Chapter 2.2: there is very little information provided how the chemical analysis has been done in detail, especially for the ion chromatography and for the ICP-AES part. E.g. I miss the information what mobile phase (e.g. which buffer) was used, which column? Was a suppressor involved? How did you calibrate?, etc. For the ICP-AES analysis, there is no explanation on how the acid digestion was conducted and how the system was calibrated. Although mentioned in the header, I find no information on quality assurance and quality control.
**Therefore, the experimental information provided is not sufficiently complete and precise to allow the reproduction by fellow scientists (ACP review criterion no.6).**

Chapter 3.1.: The evolution of the average PM2.5 concentration is presented for the years 2015-2019, however, I miss the presentation of the spread of the PM2.5 concentration of the individual filter samples. As a total of more than 800 PM2.5 samples were analysed for this study, it would be interesting to see a histogram of the PM2.5 distribution for each year, showing all PM2.5 masses of every single filter. Based on such solid statistics that is provided with >800 samples, one could also make a statement on the significance of the PM2.5 reduction from 2015 to 2019. Without the presentation of the spread of the individual sample data, I have a problem seeing a trend in the data from the years 2016-2019, and **I cannot say that the results are sufficient to support the interpretations and conclusions (ACP review criterion no.5).**

l.243-246: the information given in the text do not agree with the labelling in Figure 4. "cluster 1 (C1) included all the sites in 2015", while in Figure 4, C1 shows the years 2018 and 2019.

l. 245: How many samples are averaged behind 2016DJY3? What is the distinct pollution feature at this station? Based on Figure 4B, I cannot see a distinct chemical profile of C4, compared to the other clusters (may be a bit more ammonium, less chloride, more heavy metals?).

Chapter 3.2.1: The statements in this chapter appear to me as highly speculative. E.g. a high OC fraction does not necessarily mean that this OC only stems from fuel combustion and biomass burning. Do industrial processes not emit VOCs at layer 3? What is the role of biogenic emissions in layer 3?

Furthermore, it is stated "It's interesting to find that the spatial clustering in each year was generally consistent with the classification of three layers." – I don´t see that in Fig. 5: for example, for year 2018, cluster 2 appears in layer 3 and in layer 1; year 2017, cluster 4 appears in layer 2 and layer 3.

l. 282-284: The relative contribution of sulfate decreased, okay, but what about the evolution of the absolute concentration of sulfate, nitrate and chloride in these years? I cannot find these numbers. Maybe the relative fraction only decreases, because other compounds (e.g. secondary organics) increased from 2015-2019?

**Minor comments:**

l. 75: "A total of 836 samples were collected in 19 sites […]" → this information rather belongs to chapter 2.1.

l. 113: unit of sampling is likely L min$^{-1}$, not min L$^{-1}$. Otherwise 100 min L$^{-1}$ would be really slow.

Chapter 2.5.: A comment on the specific choice of the distance metric (cosine) and linkage function (average) would be helpful. Is the result with this particular choice robust, such that other distance and linkage functions lead to the same/ a similar result?

l. 335: "The effectiveness of the SWPSCF method was well-evaluated during the investigation." How?

l. 386: Hysplit and meteorological data are not all presented data. Please add a data repository with all presented data, or explain how to get these data.

**Technical notes:**

General: The term "layer" to describe a specific area appears to me as a bit odd. With layers I associate rather vertical stacked "layers".

l. 13 + 19 (and throughout the manuscript): better use singular of "compositions"

l. 13: "at multisite." ?? → at multiple sites.

l. 22 + 23: but dropped to less than 1.

l. 29: were occurred → occurred.

l. 45: small number of literatures → better: small number of publications

l. 61: consider rewriting the sentence "For PSCF method, due to the sources showed…". Eventually make two or three sentences out of it, for a better reception of the message.

l. 64: would be developed in this work → has been / was developed in this work, which combines […].

l. 91: "time interval of highly polluted vehicles"? You likely mean highly polluting vehicles, but what do you mean by time interval? Time interval of vehicle registration? Please specify.

l. 108: may not be fully consistent…

l. 132: The sentence is missing a verb.

l. 137: extracting → extraction.

l. 165: chemical compositions → better write "chemical species".

l.245: DJY3 is only one station → the only site far from the other sites.

l.337: are shown in Fig. 8

---

## Author Comment (AC1)

**Response to Interactive comments from Anonymous Referee #2**

Referee comments are in black. Author responses are in blue. Revised sentences are in red.

This manuscript, Spatiotemporal and policy-related variations of PM2.5 compositions and sources during 2015-2019 at multisite of a Chinese megacity, has been studied the relationship among urbanization, policy and PM2.5 components variation in a fastdeveloping Chinese city. This manuscript investigated the spatiotemporal and policyrelated variations of PM2.5 components and sources via the methods of Hierarchical Cluster Analysis and PMF, etc. Meanwhile, source weighted PSCF was developed in this work. The results and method can be useful for further policy formulation in most developing and polluted countries as well as supply basic information for future epidemiological studies. I recommend the manuscript to be published after minor corrections.

Response: I am very grateful for your comments of the manuscript. According to your advice, we amended the relevant part in manuscript. Your questions were answered below.

Some minor corrections were as followed:

(1) Latitude and longitude should be added in Figure 1 and Figure 8.

Response: Thank you very much. Figures have been revised.

---

## Author Comment (AC2)

**Response to Interactive comments from Anonymous Referee #1**

Referee comments are in black. Author responses are in blue. Revised sentences are in red.

Feng et al. present a measurement report on a $PM_{2.5}$ filter data set that covers the years 2015-2019 at ~19 stations in the urban area of Chengdu, China. This is a valuable dataset, which displays the transformation of $PM_{2.5}$ mass and composition in a rapidly developing urban area. Therefore, it is certainly within the scope of ACP. The general message is that regulations on coal combustion take effect within these five years, while $PM_{2.5}$ due to traffic emissions and secondary particles increases. This is an important finding and worth publishing. However, in the current state, I cannot recommend the manuscript being published for several reasons: (1) the manuscript requires heavy language editing. Under "technical notes" I listed a few mistakes, I could have listed more, (2) the major message of the paper is that $PM_{2.5}$ decreased between 2015 and 2019, but this is only based on the average of a large number of filter samples – no test for statistical significance (see major comment on chapter 3.1), (3) the methodology part is far from being complete – reproduction by fellow scientists would be impossible, (4) a few labels of clusters between figures and text are mixed up, (5) some of the data are interpreted in a highly speculative manner, often not convincingly supported by the data (see comment on chapter 3.2.1). Overall, I conclude that the manuscript would need a major revision in order to meet the standards of ACP.

Response: Thank you very much for your comments for the manuscript. According to the advices, we carefully proof-read the manuscript, and have made extensive modification on the original manuscript. A document answering every question from the referees was summarized. Particularly, (1) the language editing has been conducted be an English language editing service, and detailed modifications can be seen in the revised manuscript; (2) daily concentrations of total 836 $PM_{2.5}$ samples are provided in the histogram (Figure S2) to present the temporal variation in five years. We also apply the two tailed matched t-test (Table S5) to confirm the significant difference of the $PM_{2.5}$ concentrations between 2015 and 2019 (see response to major comment on

chapter 3.1); (3) the methodology part has been rewritten to meet the corresponding criterion. Specific modifications can be seen at the response to major comment on chapter 2.2; (4) the labels of clusters are corrected in text to keep it consistent with that in figures (see response to l. 243-246); (5) we have added more specific explanation in chapter 3.2.1 according to your valuable advices, and detailed information is showed in response to major comment on chapter 3.2.1.

Major comments:

l. 22: A mass ratio is not necessarily an indicator for emission reduction. What is with the absolute values of sulfate and nitrate?

Response: Thanks for the comments. We have listed the absolute concentrations of $SO_4^{2-}$ and $NO_3^-$ at each site in 2015-2019 in Fig. S4 (a) and Fig. S4 (b). Both concentrations and mass ratio are discussed for indicating the emission changes. The detailed sentence has been revised as:

The $SO_4^{2-}/NO_3^-$ mass ratio at most sites exceeded 1 in 2015 but dropped to less than 1 since 2016, reflecting decreasing coal combustion and increasing traffic impacts in Chengdu, and can be further supported by temporal variations of the $SO_4^{2-}$ and $NO_3^-$ concentrations. (Lines 22-24)

The absolute concentrations of $SO_4^{2-}$, $NO_3^-$ and $Cl^-$ are shown in Fig. S4. Publications have reported the use of $SO_4^{2-}$ and $Cl^-$ as coal-burning markers (Tian et al., 2014; Vassura et al., 2014). In the five years of the study, the average concentrations of both $SO_4^{2-}$ and $Cl^-$ sharply decreased, from 28 μg m$^{-3}$ to 8 μg m$^{-3}$ and from 6 μg m$^{-3}$ to 2 μg m$^{-3}$, respectively. The fractions of $SO_4^{2-}$ and $Cl^-$ also showed a decreasing trend, especially in 2016. However, the fractions of $NO_3^-$ showed a general increasing trend from 2015 to 2019. The average concentrations of $NO_3^-$ were found to decrease from 20 μg m$^{-3}$ in 2015 to 14 μg m$^{-3}$ in 2016, mainly resulting from the strongly promoted coal-burning ban policy, after that, $NO_3^-$ increased slightly to 16 μg m$^{-3}$ in 2019, which might be attributed to the gradually enhanced contribution of vehicles and use of natural gas. We also analyzed the $SO_4^{2-}/NO_3^-$ mass ratio, a qualitative indicator of sulfur versus nitrogen sources (Gao et al., 2015; Arimoto et al., 1996), and the summary is presented

in Fig. S4 (d). Ratios at most sites exceeded 1 in 2015, dropped to less than 1 in 2016, and then declined steadily. Combined with the absolute concentrations of $SO_4^{2-}$ and $NO_3^-$ discussed above, the $SO_4^{2-}/NO_3^-$ mass ratio can also indicate decreasing coal combustion and increasing traffic emissions in Chengdu. (Lines 333-344)

l. 25: A large fraction of secondary particle → this cannot be attributed to individual sources, such as traffic, coal combustion, industrial emissions? E.g., secondary sulfate can partly originate from coal combustion.

Response: Thank you very much for this professional and helpful comment. According to the principle of the PMF receptor model, it extracts factors based upon their different chemical profiles and temporal variations. The secondary particles were not directly emitted by primary sources, but were formed through atmospheric processes. Thus, primary sources and secondary particles are separately identified in most studies using receptor models. Furthermore, the relationships between secondary particles and individual sources were qualitatively discussed in this work. For example, the decreased $SO_4^{2-}$ was closely related to the reduction in coal combustion, the high level of $NO_3^-$ may indicate the increasingly serious traffic emissions, and the renovation of de-nitrification in industries would be a potential reason for the declining level of $NO_3^-$. (Lines 333-344)

l. 28: What do you mean by "stronger distribution patterns."?

Response: In Chapter 3.3.1, the CV (Coefficient of Variation) was calculated to investigate the spatial difference of each source category. The high CV values for coal and biomass combustion source and industrial emission source indicate they showed stronger spatial difference than other sources. The confused sentence has been revised as: "For spatial variations, the high coefficient of variation (CV) values of coal and biomass combustion and industrial emissions indicated their higher spatial difference in Chengdu." (Lines 28-29)

Chapter 2.2: there is very little information provided how the chemical analysis has been done in detail, especially for the ion chromatography and for the ICP-AES part. E.g., I miss the information what mobile phase (e.g., which buffer) was used, which column? Was a suppressor involved? How did you calibrate?, etc. For the ICP-AES analysis, there is no explanation on how the acid digestion was conducted and how the system was calibrated. Although mentioned in the header, I find no information on quality assurance and quality control. Therefore, the experimental information provided is not sufficiently complete and precise to allow the reproduction by fellow scientists (ACP review criterion No.6).

Response: Thanks for the suggestion. We rewrite the chapter 2.2 and provide the detailed experimental information in order to meet the ACP review criterion No.6. Questions you asked above are answered in Lines 126-173 and Table S3:

**2.2 Chemical analysis and quality assurance/ quality control (QA/QC)**

[revised manuscript text omitted]

Chapter 3.1.: The evolution of the average $PM_{2.5}$ concentration is presented for the years 2015-2019, however, I miss the presentation of the spread of the $PM_{2.5}$ concentration of the individual filter samples. As a total of more than 800 $PM_{2.5}$ samples were analysed for this study, it would be interesting to see a histogram of the $PM_{2.5}$ distribution for each year, showing all $PM_{2.5}$ masses of every single filter. Based on such solid statistics that is provided with >800 samples, one could also make a statement on the significance of the $PM_{2.5}$ reduction from 2015 to 2019. Without the presentation of the spread of the individual sample data, I have a problem seeing a trend in the data from the years 2016-2019, and I cannot say that the results are sufficient to support the interpretations and conclusions (ACP review criterion no.5).

Response: Thank you very much. In Figure S2, daily PM$_{2.5}$ concentrations and annual average PM$_{2.5}$ concentrations are provided based on sampling date at each site. We map the histograms and line charts to present the temporal variation of PM$_{2.5}$ from 2015 to 2019. Furthermore, the two tailed matched t-tests are applied and the result summarized in Table S5, which indicated the significant difference between 2015 and 2019:

Figure S2 shows the temporal variation of daily PM$_{2.5}$ concentrations and annual average PM$_{2.5}$ concentrations for each site. The large number of sampling data from all filters further demonstrates the temporal changes in PM$_{2.5}$ concentrations over time, as described above. The results of the statistical analysis, using the two tailed matched t-tests for PM$_{2.5}$ concentrations at sampling sites between 2015 and 2019, are summarized in Table S5. As seen in the table, there was a significant decreasing trend in the level of PM$_{2.5}$ in the period 2015-2019. (Lines 263-267)

l.243-246: the information given in the text do not agree with the labelling in Figure 4. "cluster 1 (C1) included all the sites in 2015", while in Figure 4, C1 shows the years 2018 and 2019.

Response: Thank you very much. The confused information has been revised: Four clusters were identified, and the results showed a strong correlation with years: Cluster 1 (C1) consisted of most sites in 2018 and 2019; sites in 2016 and 2017 were classified as cluster 2 (C2); cluster 3 (C3) included all the sites in 2015; and 2016DJY3, the only site far from the other sites, was separated as cluster 4 (C4). (Lines 284-286)

l. 245: How many samples are averaged behind 2016DJY3? What is the distinct pollution feature at this station? Based on Figure 4B, I cannot see a distinct chemical profile of C4, compared to the other clusters (may be a bit more ammonium, less chloride, more heavy metals?).

Response: Thanks for the comment. We add the related explanation as: "A total of thirteen samples were collected at 2016DJY3, and both the sampling number and duration were similar to samples collected at other sites in 2016. As a typical background site in Chengdu, DJY3 is surrounded by plants and agricultural activities,

so it is featured by the distinctive $PM_{2.5}$ compositions with markedly high $NH_4^+$ and crustal elements. This explains the particular HCA result of the C4 well." (Lines 286-290)

Chapter 3.2.1: The statements in this chapter appear to me as highly speculative. E.g. a high OC fraction does not necessarily mean that this OC only stems from fuel combustion and biomass burning. Do industrial processes not emit VOCs at layer 3? What is the role of biogenic emissions in layer 3? Furthermore, it is stated "It's interesting to find that the spatial clustering in each year was generally consistent with the classification of three layers." – I don´t see that in Fig. 5: for example, for year 2018, cluster 2 appears in layer 3 and in layer 1; year 2017, cluster 4 appears in layer 2 and layer 3.

Response: Thank you very much for this helpful and professional suggestion. Firstly, it's right to consider the VOC emission from industrial process. We consider both the primary organic carbon (POC) and the secondary organic carbon (SOC) to explain the high OC levels. With respect to SOC, we have studied related publications about VOCs source profiles and conclude the specific VOCs sources during the industrial process. The corresponding manuscript is revised as: "The higher OC fractions of these clusters were considered to occur at the contained sites, such as PZ3, JT3, CZ3, XJ3, JY3, and PJ3, and could be either directly emitted (primary organic carbon, POC) or indirectly formed in the atmospheric (secondary organic carbon, SOC) (Kanakidou et al., 2005; Zhong et al., 2021). The high POC was largely associated with the stronger fuel combustion and biomass burning. One possible reason is that there were more residential combustions (such as bulk coal and biofuel combustion) and small boilers with low combustion efficiency at PZ3 and XJ3; therefore, control measures for fuel combustion still need to be strengthened. During the sampling period, activities such as the burning of firewood by residents to produce smoked meat can contribute greatly to the OC level from biomass burning. Additionally, the formation of SOC was also responsible for the high OC level. SOC is generated from the oxidation of volatile organic compounds (VOCs) through homogeneous or heterogeneous reactions (Jang et

al., 2002). VOC precursors come from both anthropogenic sources and plant emissions (Ait-Helal et al., 2014; Kleindienst et al., 2009). Previous studies (Zhao et al., 2018; Han et al., 2013; Yin et al., 2015) have reported high VOC emissions from industrial processes at PZ3, JT3, and other sites. Coal combustion in industries and thermal power plants were the main sources of industrial processes at PZ3 and JT3, respectively." (Lines 305-317) Secondly, we add the discussion about the biogenic emissions from VOCs, related manuscript is revised as: "Biogenic VOC emissions often occur at several agriculture sites such as JY3 and PJ3 because of the high vegetation coverage in these areas." (Lines 317-318) Thirdly, the sentence that "It's interesting to find that the spatial clustering in each year was generally consistent with the classification of three layers" is inappropriate and we delete it. Instead, we focus the characteristics of the specific sampling site when discussing the spatial variations of chemical compositions in Chapter 3.2.1, the manuscript is revised as: "Spatial differences were observed each year. The clusters 2015C4, 2016C4, 2017C1, 2018C2, and 2019C1 always showed higher OC fractions: 20.9%, 14.6%, 20.5%, 17.5%, and 23.3% of $PM_{2.5}$ mass, respectively. The higher OC fractions of these clusters were considered to occur at the contained sites, such as PZ3, JT3, CZ3, XJ3, JY3, and PJ3, and could be either directly emitted (primary organic carbon, POC) or indirectly formed in the atmospheric (secondary organic carbon, SOC) (Kanakidou et al., 2005; Zhong et al., 2021)." (Lines 304-308)

l. 282-284: The relative contribution of sulfate decreased, okay, but what about the evolution of the absolute concentration of sulfate, nitrate and chloride in these years? I cannot find these numbers. Maybe the relative fraction only decreases, because other compounds (e.g. secondary organics) increased from 2015-2019?

Response: Thanks for the suggestion. Similar to the response to l. 22, the absolute concentration of sulfate, nitrate and chloride are presented in Figure S4. We discussed the temporal variations of them based on both concentration and fraction: "The absolute concentrations of $SO_4^{2-}$, $NO_3^-$ and $Cl^-$ are shown in Figure S4. Publications have reported the use of $SO_4^{2-}$ and $Cl^-$ as coal-burning markers (Tian et al., 2014; Vassura et

al., 2014). In the five years of the study, the average concentrations of both $SO_4^{2-}$ and $Cl^-$ sharply decreased, from 28 µg m$^{-3}$ to 8 µg m$^{-3}$ and from 6 µg m$^{-3}$ to 2 µg m$^{-3}$, respectively. The fractions of $SO_4^{2-}$ and $Cl^-$ also showed a decreasing trend, especially in 2016. However, the fractions of $NO_3^-$ showed a general increasing trend from 2015 to 2019. The average concentrations of $NO_3^-$ were found to decrease from 20 µg m$^{-3}$ in 2015 to 14 µg m$^{-3}$ in 2016, mainly resulting from the strongly promoted coal-burning ban policy, after that, $NO_3^-$ increased slightly to 16 µg m$^{-3}$ in 2019, which might be attributed to the gradually enhanced contribution of vehicles and use of natural gas." (Lines 333-340) The result indicates the decreasing trend for both concentration and fraction.

Minor comments:

l. 75: "A total of 836 samples were collected in 19 sites […]" → this information rather belongs to chapter 2.1.

Response: Thank you. We delete this sentence in the wrong place. The corresponding sentence is revised at Line 112.

l. 113: unit of sampling is likely L min$^{-1}$, not min L$^{-1}$. Otherwise 100 min L$^{-1}$ would be really slow.

Response: Thank you. It has been revised. (Line 115)

Chapter 2.5.: A comment on the specific choice of the distance metric (cosine) and linkage function (average) would be helpful. Is the result with this particular choice robust, such that other distance and linkage functions lead to the same/ a similar result?

Response: Thank you. We add the related explanation in the Supplement (Lines 15-27):

"**2.5 Hierarchical cluster analysis (HCA)**

A selection of the distance metric and linkage method is required during hierarchical clustering. Common distance metrics include Euclidean, cosine, Manhattan,

Minkowski, and Hamming distances. The cosine distance is particularly well-suited to describing the relationship between objects, given that it places more emphasis on the relative difference in direction and is not sensitive to absolute value. In this study, the selection of the distance metrics was found to have little influence on the result. Among the linkage methods, the single, complete, average, centroid, and Ward linkage are commonly used methods. The average linkage method calculates the average distance between groups rather than the minimum and maximum distances, avoiding the shortcomings of single and complete linkages. The application of the Ward linkage method is based on the precondition that the Euclidean distance is measured (Kalkstein et al., 1987; Chernoff, 1975; Ward, 1963) (Chernoff, 1975; Kalkstein et al., 1987; Ward, 1963). To guarantee the effectiveness of the algorithm, the cosine distance and average linkage method was ultimately selected for the cluster analysing."

And the manuscript is revised as:

"To guarantee the effectiveness of the algorithm, appropriate methods should be selected according to the properties of specific objects. An introduction on the specific choice of the distance metric and linkage function is added in the Supplement. In this study, the HCA was conducted using IBM SPSS Statistics 25, and the results were confirmed to be similar by using different distance metrics and linkage methods. Based on the comprehensive consideration, the HCA based on the cosine distance and average linkage method was selected." (Lines 242-246)

l. 335: "The effectiveness of the SWPSCF method was well-evaluated during the investigation." How?

Response: Thank you very much. This sentence was put at the wrong place, resulting in the unclear expression. We delete the sentence at the original place and add the concluding sentences at the end of the paragraph: "As described above, the potential source regions identified after the source weighting could better reflect the spatial variations of source contributions, suggesting the effectiveness of the SWPSCF method in this study." (Lines 398-400)

l. 386: Hysplit and meteorological data are not all presented data. Please add a data repository with all presented data, or explain how to get these data.

Response: Thanks for the suggestion. We explain the source of data. Related sentences are revised as: "In this study, the backward trajectories were modelled using the MeteoInfo, a GIS application which enables the user to visualize and analyse the spatial and meteorological data with multiple data formats, which is available at http://www.meteothinker.com/. The required meteorological data were obtained from the National Centers for Environmental Prediction (NCEP) global reanalysis data, which are available from the National Oceanic and Atmospheric Administration (NOAA's) Air Resources Laboratory (ARL) in a format suitable for transport and dispersion calculations. A detailed dataset can be obtained from the NOAA ARL FTP server (https://ready.arl.noaa.gov/archives.php)." (Lines 198-203) Should you have any questions, please contact us without hesitate.

Technical notes:

General: The term "layer" to describe a specific area appears to me as a bit odd. With layers I associate rather vertical stacked "layers".

Response: We use the term "layer" for the reason that Chengdu residents are accustomed to defining different zones as "layers" according to the division of the specific circle road. Corresponding explanation of "layers" can be seen in Chapter 2.1 (Lines 96-108).

l. 13 + 19 (and throughout the manuscript): a better use singular of "compositions"

Response: Thanks. They are revised throughout the manuscript.

l. 13: "at multisite." ?? → at multiple sites.

Response: Thanks. They are revised. (Lines 13-14)

l. 22 + 23: but dropped to less than 1.

Response: Thanks. It is revised. (Line 22)

l. 29: were occurred → occurred.

Response: Thanks. It is revised. (Line 29)

l. 45: small number of literatures → better: small number of publications

Response: Thanks. It is revised. (Lines 46-47)

l. 61: consider rewriting the sentence "For PSCF method, due to the sources showed…". Eventually make two or three sentences out of it, for a better reception of the message.

Response: Thanks. The corresponding sentence is revised as: "In the traditional PSCF method, the source localization is mainly based on the number of trajectory endpoints that fall in the targeted grid cell. However, it should not be ignored that the sources showed discrepant spatial distribution patterns over the studied region. When trajectories pass over the grid cell in which a source category shows high local contributions, the probability of potential contribution for this grid cell should theoretically be relatively high." (Lines 62-66)

l. 64: would be developed in this work → has been / was developed in this work, which combines […].

Response: Thanks. The sentence is revised as: "Accordingly, we developed a source weighted PSCF (SWPSCF) method that combines PMF with PSCF and considers the spatial distribution of contributions for each source category." (Lines 66-67)

l. 91: "time interval of highly polluted vehicles"? You likely mean highly polluting vehicles, but what do you mean by time interval? Time interval of vehicle registration? Please specify.

Response: Thanks. The related sentence is revised as: "To improve air quality, the Chengdu government adopted several measures, including limiting the driving area of

highly polluting vehicles and setting specific hours for driving in, adjusting industrial structures, and implementing energy substitution." (Lines 91-93)

l. 108: may not be fully consistent…

Response: Thanks. It is revised. (Line 111)

l. 132: The sentence is missing a verb.

Response: Thanks. The sentence is deleted during rewriting the Chapter 2.2. The corresponding sentence is modified as: "For QA/QC, a system stability test (three-peak detection) is required before and after detecting samples and the relative standard deviation should not exceed 5%." (Lines 142-143)

l. 137: extracting → extraction.

Response: Thanks. It is revised. (Line 146)

l. 165: chemical compositions → better write "chemical species".

Response: Thanks. They are revised throughout the manuscript.

l.245: DJY3 is only one station → the only site far from the other sites.

Response: Thanks. It is revised. (Line 286)

l.337: are shown in Fig. 8

Response: Thanks. It is revised. (Line 392)